# SIGNED-DICTIONARY AND NONNEGATIVE-ACTIVATION DECOMPOSITION FOR CONCEPT BOTTLENECK MODELS

## ABSTRACT

Concept-driven interpretability often relies on a fixed text pool, which limits coverage of fine-grained and compositional concepts and weakens the coupling between explanations and decisions. We introduce **SAND-CBM**, a label-free framework that learns concepts directly from image representations in an aligned vision–language space. SAND-CBM factorizes features into a *signed* concept dictionary $W$ and *nonnegative* activations $U$, then applies a scale-equivalent normalization that maps each activation column to $[0, 1]$ for comparable strength across concepts. A class-conditional sparse gate enables per-class selection over a shared dictionary, supporting reuse without per-class redundancy. On top of the same $(U, W)$, we expose two lightweight and complementary usage modes: *Branch-A* concatenates image–text similarities with $U$ in a CBM-style interface, while *Branch-B* concatenates back-mapped reconstructions $Z=UW^*$ with $U$ in a CEM-style interface. Across CIFAR–100, CUB, and SUN, SAND-CBM attains 80.52%, 80.76%, and 67.64% Acc@1, respectively, yielding an average gain of 10.14% over all baselines. Our code is available at: https://anonymous.4open.science/r/SAND-FA73/.

## 1 INTRODUCTION

Concept-driven interpretable learning—exemplified by *Concept Bottleneck Models* (CBM) (Koh et al., 2020) and *Concept Embedding Models* (CEM) (Espinosa Zarlenga et al., 2022)—adopts a "first recognize concepts, then decide" paradigm, exposing a clear human-semantic interface. However, in *label-free* settings (Oikarinen et al., 2023), existing approaches typically rely solely on a pre-specified text pool: they compute similarities between image representations and a set of textual concept embeddings to obtain "concept scores." While simple to implement, this paradigm is intrinsically constrained by the incompleteness and bias of the text pool—human-curated concept sets struggle to cover fine-grained and compositional semantics; long-tail and cross-domain concepts are even harder to exhaust—leading to a cascade of issues: missing concepts, weak coupling between explanations and decisions, and capped performance. Moreover, discretizing continuous semantics (e.g., "red hair, light red, dark red") into single a text embedding induces information loss, making it difficult for the explanatory layer to faithfully capture subtle yet rich visual factors. In contrast, image-side signals are denser and more fine-grained—one core reason current label-free pipelines remain fundamentally limited.

Within label-free CBM variants, both post-hoc and residual styles suffer structural limitations. *Post-hoc* routes (Yuksekgonul et al., 2023) often attach concept heads or residual branches to a pretrained black box to "match" its decisions, yet the newly learned channels need not align with crisp, nameable concepts, casting doubt on interpretability. *CF-CBM* (Dominici et al., 2024) introduces hierarchical concept organization but depends on costly ontology annotations and yields high-level features with limited specificity. *Res-CBM* (Shang et al., 2024) injects a small fixed quota of residual concepts to recover accuracy, but real-world compositional semantics far exceed any preset quota, inevitably missing key factors. More fundamentally, many methods do not truly *extract* concepts from images; instead they *project* onto an existing textual coordinate system, rendering the explanatory space governed by text coverage rather than by the data's own interpretable structure.

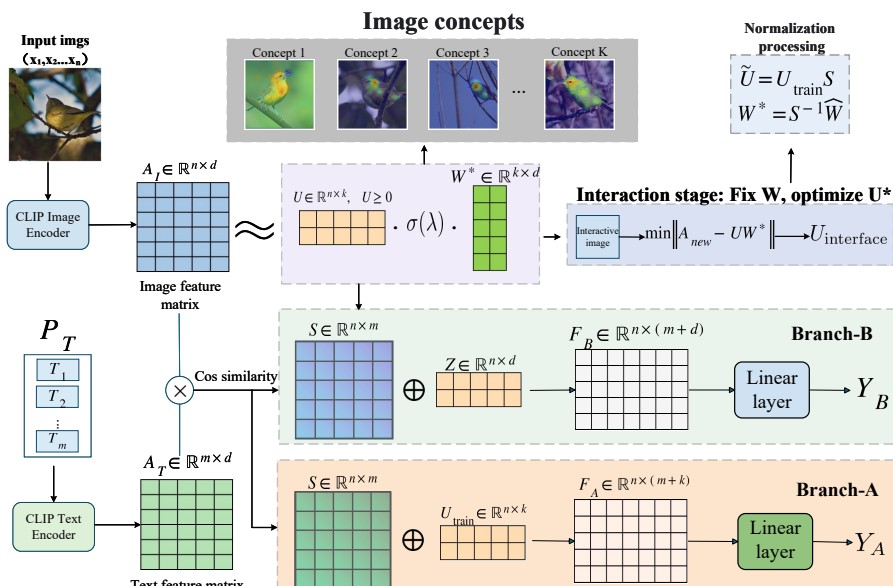

Figure 1: **Overview of SAND-CBM.** CLIP encodes images (a text pool) into features, which are factorized into a signed concept dictionary and nonnegative activations with class-conditional gating and normalization. Predictions use two lightweight branches: Branch-A concatenates image–text similarity with activations, and Branch-B concatenates back-mapped concept reconstructions with activations. Design and implementation notes and theoretical summary, please refer to Appendix A.

Classical concept extraction via Nonnegative Matrix Factorization (NMF) (Lee & Seung, 1999; Fel et al., 2023) is also misaligned with modern aligned representations (e.g., CLIP (Radford et al., 2021)) in two ways: (i) NMF constrains both the dictionary and activations to be nonnegative, which conflicts with *signed* semantic axes in aligned spaces (e.g., bright–dark, rough–smooth); even applying a bias shift to maintain nonnegativity inflates vector similarities and disrupts directional information. (ii) Many implementations learn concepts *per class*, hindering cross-class sharing, inducing redundancy, impeding transfer, and running counter to open-vocabulary needs.

To systematically address these bottlenecks, we propose **SAND-CBM** (Signed-dictionary And Nonnegative-activation Decomposition). As illustrated in Fig. 1, within CLIP's visual–semantic space we directly and unsupervisedly learn a *signed concept dictionary* $W$ and *nonnegative concept activations* $U$, together with *class-conditional sparse gating* that enables per-class selection over a shared large dictionary. Nonnegative $U$ endows an additive and comparable notion of concept *strength*; signed $W$ respects the bipolar semantics of aligned spaces. We further adopt *scale-equivalent normalization* to map each column of $U$ to $[0, 1]$, establishing a unified strength scale across concepts without sacrificing the optimal reconstruction. On the same $(U, W)$, SAND-CBM offers two lightweight, complementary usage branches: **Branch-A (CBM-based)** constructs an image–text similarity matrix $S$ and concatenates it with $U$ as $[S \mid U]$, preserving interface compatibility and auditability with textual concepts; **Branch-B (CEM-based)** first maps concepts back to the CLIP semantic space via $Z = UW$, then concatenates with $U$ as $[S \mid Z]$ to exploit the discriminative expressivity of image-side concept synthesis. This unified design both relaxes rigid dependence on text-pool coverage and naturally interoperates with CBM/CEM interfaces, facilitating deployment and drop-in replacement.

Our main contributions are: **(1) Image-side concept discovery:** directly learning concept dictionaries and activations in an aligned semantic space so that fine-grained and compositional semantics are no longer bounded by manual enumeration; **(2) A signed-dictionary + nonnegative-activation factorization:** decoupling "direction" ($W$) from "strength" ($U$), with scale-equivalent normalization that bounds activations in $[0, 1]$; **(3) Unified modeling of class sharing with per-class selection:** class-conditional sparse gating selects subsets from a shared dictionary, avoiding per-class redundancy and improving transfer; **(4) A dual-branch usage scheme compatible with CBM/CEM interfaces:**

retaining text alignment as an explanatory anchor (Branch-A) while strengthening discriminative power via reconstruction back-mapping (Branch-B), yielding complementary benefits across concept budgets and task types; **(5) Consistent empirical gains with structural diagnostics:** across multiple datasets we observe stable improvements over representative baselines, and correlation/gating visualizations support an *atomic, low-redundancy, compositional* organization of concepts.

## 2 RELATED WORK

Research on *interpretable, concept-driven prediction* has largely progressed along two threads: (i) *how concepts are used for decision-making*—employing concepts as intermediate representations or embedded features to drive classification (e.g., *Concept Bottleneck Models* (CBM) (Koh et al., 2020) and *Concept Embedding Models* (CEM) (Espinosa Zarlenga et al., 2022)); and (ii) *how concepts are obtained and completed*—evolving from explicit concept supervision toward label-free discovery, post-hoc extraction, incremental residual completion, and coarse-to-fine hierarchical refinement (e.g., LF-CBM (Oikarinen et al., 2023), PH-CBM (Yuksekgonul et al., 2023), Res-CBM (Shang et al., 2024), CF-CBM (Dominici et al., 2024)). Yet under open-vocabulary and distributionally diverse settings, several issues remain salient: incomplete text pools, incomparable concept scales, weak coupling between explanation and decision, and heavy reliance on quota or hierarchical priors. Our method, *SAND-CBM* (§3.3– 3.5), addresses these bottlenecks from the *image side* via a unified framework of a *signed concept dictionary $W$*, *nonnegative activations $U$*, and *class-conditional sparse gating*.

**CBM.** CBM (Koh et al., 2020) decomposes prediction into "concept recognition $\rightarrow$ label prediction," providing *intervenability* and *auditability*. However, when the concept set is incomplete or concept predictions are noisy, errors *cascade* through the bottleneck to the final label, inducing a performance–interpretability trade-off. In §3.5, Branch-A adopts a *usage pattern* that *concatenates text similarity $C$* with *concept activations $U$*: with $A_T = E_T(P_T)$ and $C = A_I A_T^\top (= S)$, this preserves *interface compatibility* with the CBM style rather than engaging in a closed-form performance competition with the CBM family.

**CEM.** CEM (Espinosa Zarlenga et al., 2022) aligns concept embeddings with high-dimensional model representations, alleviating the information bottleneck of "hard" CBM interfaces; nevertheless, alignment quality still hinges on the coverage and fidelity of a pre-defined concept set. In §3.5, Branch-B *concatenates concept reconstruction $Z = UW^*$* with *activations $U$*, functionally mirroring CEM-style alignment/reconstruction as a *usage paradigm* rather than a direct objective.

In label-free and post-hoc concept completion, we contrast four routes in a unified view. **LF-CBM** (Oikarinen et al., 2023) reduces labeling by leveraging weak inductive biases and self-supervised signals together with prompts, prototypes, and clustering, but suffers from *incompleteness* (text-pool coverage misses fine-grained and compositional long-tail concepts) and *non-atomicity* (entangled pseudo-concepts). **PH-CBM** (Yuksekgonul et al., 2023) attaches concept heads *post hoc*, which is model-agnostic yet decouples explanations from decision boundaries and introduces auxiliary paths with unclear semantics. **Res-CBM** (Shang et al., 2024) adds residual concept channels under a fixed budget to close accuracy gaps, but quotas miss diverse compositions and residual semantics may drift across tasks. **CF-CBM** (Dominici et al., 2024) imposes hierarchical concepts, incurring costly ontologies with poor cross-domain transfer; hierarchies can overlap, emphasize textual and ontological design over image-side extraction, and rely on high-level features that are weak for explanation. In contrast, **SAND-CBM** learns *from the image side* a *signed* dictionary $W$ with *nonnegative* activations $U$, applies a *scale-equivalent* normalization mapping $U$ to $[0, 1]$ for comparable strength, and uses *class-conditional sparse gating* to select per-class subsets from a *shared* dictionary (§3.3, 3.4); integrating factorization and gating end to end aligns explanatory pivots with decisions, low-correlation regularization on $W$ together with the additivity of $U$ promotes atomic, low-redundancy concepts, and avoiding small-patch extraction reduces distributional mismatch with CLIP pretraining while enabling coarse-to-fine narrowing without explicit hierarchical labels.

For concept discovery and alignment, TCAV (Kim et al., 2018), ACE (Ghorbani et al., 2019), and prototype networks (e.g., ProtoPNet (Chen et al., 2019)) align *human semantics* with model representations for sensitivity analysis or prototype matching, but typically rely on explicit concept sets and text, and place less emphasis on *additivity and scale comparability*. Meanwhile, some methods (e.g., CF-CBM (Dominici et al., 2024)) construct concepts per class without *sharing*, which

is incompatible with our "unified dictionary + class-conditional gating" setting and thus not directly applicable for a fair comparison. On interpretable factorization, NMF, dictionary learning, and sparse coding (Lee & Seung, 1999; Aharon et al., 2006) promote part-based, additive explanations. We extend this line into CLIP's *signed* semantic space: *nonnegative* activations $U$ preserves *additivity*, the *signed* dictionary $W$ matches directional structure in CLIP, and *scale equivalence* establishes a unified strength coordinate (§3.3).

## 3 METHOD

### 3.1 NOTATION AND PROBLEM SETUP

**Definition of CBM.**   A CBM can be formalized as a pair $(h, f)$ consisting of a *concept extractor* $h : X \to \mathbb{R}^k$ and a *classifier* $f : \mathbb{R}^k \to \mathcal{Y}$. In our framework, $h$ operates on the CLIP (Radford et al., 2021) representation of an input $x$ and returns concept values via similarities to pre-specified text vectors; $f$ then predicts a label $y \in \mathcal{Y}$ from these concept values.

**Data and Encoders.**   Given images $X = \{x_1, \ldots, x_n\}$, the CLIP image encoder $E_I(\cdot)$ yields a $d$-dimensional feature matrix $A_I = E_I(X) = \begin{bmatrix} a_1^\top \\ \vdots \\ a_n^\top \end{bmatrix} \in \mathbb{R}^{n \times d}$, where $d = 512$ for the default ViT-B/16 backbone (Dosovitskiy et al., 2021) (and changes accordingly for other backbones). The $i$-th row $a_i$ is the $d$-dimensional CLIP representation of $x_i$. Notably, for computational convenience and stable similarity computation, the output of $E_I(\cdot)$ is L2-normalized, i.e., $a_i \leftarrow a_i / \|a_i\|_2$. Thanks to CLIP's vision–language alignment, $A_I$ and text features lie in a shared semantic space, enabling textual "probes" to retrieve and name concepts in images.

**Concept Factorization and Constraints.**   Within the CLIP representation space we learn a *concept activation matrix* and a *concept dictionary*: $U = \begin{bmatrix} u_1^\top \\ \vdots \\ u_n^\top \end{bmatrix} \in \mathbb{R}_{\geq 0}^{n \times k}, \qquad W \in \mathbb{R}^{k \times d}$, where $k$ is the number of concepts (a hyperparameter). Each $u_i \in \mathbb{R}_{\geq 0}^k$ encodes the nonnegative strength of $k$ concepts for sample $x_i$, while each row of $W$ parameterizes a concept as a *signed* direction/weight vector in CLIP's semantic space (allowing positive/negative entries to respect CLIP's bipolar axes).

**Optional Text Pool and Vision–Language Similarity.**   When incorporating language concepts, we encode a pre-specified text set $P_T = \{T_1, \ldots, T_m\}$ (Dominici et al., 2024) via the CLIP text encoder: $A_T = E_T(P_T) \in \mathbb{R}^{m \times d}$, and compute the ($L_2$-normalized dot-product) image–text similarity matrix $S = A_I A_T^\top \in \mathbb{R}^{n \times m}$. Because images and texts share a semantic coordinate system, $S$ serves as a concept signal that can complement the semantics captured by $h(x)$.

**Classes and Gating.**   For $c$-way classification, we enable cross-class concept sharing with within-class selectivity via class-conditional gating: $\lambda \in \mathbb{R}^{c \times k}, \qquad g_y = \sigma(\lambda_y) \in (0, 1)^k$, where $\sigma(\cdot)$ is the elementwise sigmoid and $g_y$ is a (sparse) selection vector for class $y$. For a labeled sample $(x_i, y_i)$, gating on the activation side realizes "shared dictionary + class-specific sparse subsets," which plugs naturally into a downstream linear head to complete the "concept $\to$ label" mapping.

### 3.2 MOTIVATION AND OVERVIEW

Existing label-free and zero-shot CBM pipelines (Oikarinen et al., 2023) rely **only** on a human-curated text pool $P_T$, which leads to *incomplete* concepts and missing fine-grained/compositional constructs. Concretely, if $\mathcal{K}$ denotes the latent universe of concepts but the text-induced set is a strict subset $\mathcal{K}_T \subsetneq \mathcal{K}$, then many true concepts cannot be enumerated or named. A toy example on CUB: the pool may include `red beak`, `black head`, `white tail`, but omit finer concepts like `white tail-tip` or conjunctive ones like `red beak & black head`, degrading zero-shot alignment to the concepts present in the image. To mitigate this, we **learn from the image side** an unsupervised *concept dictionary* $W$ together with *nonnegative activations* $U$ in CLIP's visual

space (Radford et al., 2021), producing reusable and interpretable concept representations; we then perform **class-conditional sparse gating** to select class-specific subsets from a unified dictionary. Finally, we instantiate two unsupervised branches that mirror CBM- and CEM-style usage.

**Label-free CBM with CLIP: a baseline.** In the traditional label-free setup, one first encodes $P_T$ to obtain $A_T = E_T(P_T) \in \mathbb{R}^{m \times d}$ (Radford et al., 2021). For any image $x$, we get image feature $A_I = E_I(x)$, the dot product yields a *text-driven concept vector* $h(x) = A_I A_T^\top \in \mathbb{R}^m$, which feeds a linear (or zero-shot) classifier $f$. This baseline is simple but its expressivity is entirely bounded by $P_T$: if $P_T$ does not cover $\mathcal{K}$ (e.g., missing `white tail-tip` or the conjunction `red beak & black head`), $h(x)$ cannot represent concepts that truly exist in the image. We therefore introduce *image-side concept extraction* to address incompleteness.

### 3.3 IMAGE-SIDE CONCEPT EXTRACTION

Compared to a limited text pool, images carry richer fine-grained and compositional signals. We therefore mine latent concepts directly from *image representations*, avoiding hard dependence on text coverage (Olshausen & Field, 1996; Aharon et al., 2006; Mairal et al., 2010; Tibshirani, 1996). Classical NMF (Lee & Seung, 1999) constrains $U \geq 0$ and $W \geq 0$, which (i) cannot capture bipolar semantics on the same CLIP axis (e.g., bright–dark), and (ii) often leads to *per-class* concept learning with weak cross-class sharing. We instead impose nonnegativity only on $U$ (to preserve additive *strength* semantics), while *allowing $W$ to be signed* so as to fully exploit CLIP's directional structure.

**Base Factorization and Training.** In **SAND-CBM**, we decompose $A_I = E_I(X_{\text{train}})$ from the training set into a nonnegative activation matrix $U$ and a concept dictionary $W$ by minimizing the reconstruction loss

$$U_{\text{train}}, \hat{W} = \min_{U \geq 0, W} \frac{1}{n} \left\| A_I - UW \right\|_F^2, \tag{1}$$

yielding $U_{\text{train}} \in \mathbb{R}_{\geq 0}^{n \times k}$ and $\hat{W} \in \mathbb{R}^{k \times d}$. Nonnegativity on $U_{\text{train}}$ ensures *additive* concept strengths:

$$a_i \approx \sum_{j=1}^{k} u_{ij} w_j, \qquad u_{ij} \geq 0, \tag{2}$$

where $u_{ij}$ quantifies the nonnegative "activation strength" of concept $j$ on sample $i$. The dictionary $\hat{W}$ is *unconstrained in sign*, enabling bipolar semantics in CLIP space. A toy illustration: suppose $w_1$ encodes "bright" and $w_2$ encodes "dark", $w_1 = \begin{bmatrix} 1 \\ 0.5 \end{bmatrix}$, $w_2 = \begin{bmatrix} -0.8 \\ 0.3 \end{bmatrix}$, $u_i = \begin{bmatrix} 0.6, & 0.2 \end{bmatrix}$, then $a_i \approx 0.6\,w_1 + 0.2\,w_2 = \begin{bmatrix} 0.6 \\ 0.3 \end{bmatrix} + \begin{bmatrix} -0.16 \\ 0.06 \end{bmatrix} = \begin{bmatrix} 0.44 \\ 0.36 \end{bmatrix}$. Here the activation $u_i$ is *additive* (0.6 for "bright," 0.2 for "dark"), while the signed $w_2$ introduces a negative contribution along one feature dimension, capturing bipolarity. Thus $u_{ij}$ remains an intuitive nonnegative *strength*, and $w_j$'s signs encode opposing semantic directions. A training-time complexity analysis of Eq. (1) is provided in Appendix B.

**Normalization via Scale-Equivalent Transformation (0–1 Comparability).** Concept columns may differ in scale, complicating cross-concept comparison. We rescale each activation column to $[0, 1]$ using

$$s_j = \frac{1}{\max(\|U_{\text{train}}(:, j)\|_\infty, \varepsilon)}, \qquad S = \text{diag}(s_1, \ldots, s_k) \in \mathbb{R}^{k \times k}, \tag{3}$$

with a small $\varepsilon > 0$ to avoid division by zero. Since $S$ is diagonal, $S^{-1} = \text{diag}(1/s_1, \ldots, 1/s_k)$ is trivial to compute.

We then apply the scale-equivalent transform

$$\tilde{U} = U_{\text{train}} S \in \mathbb{R}_{\geq 0}^{n \times k}, \qquad W^* = S^{-1} \hat{W} \in \mathbb{R}^{k \times d}, \tag{4}$$

which preserves reconstruction:

$$A_I = U_{\text{train}} \hat{W} = (U_{\text{train}} S)(S^{-1} \hat{W}) = \tilde{U} W^*. \tag{5}$$

By construction $\|\tilde{U}(:,j)\|_\infty = 1$, so each activation column lies in $[0,1]$. We henceforth write $U$ for the normalized activations ($U \leftarrow \tilde{U}$). For example, if a sample's raw activations are $(3.0, 0.8)$ and the training-set columnwise maxima are $(5.0, 0.8)$, then normalization by $S = \mathrm{diag}(1/5.0, 1/0.8)$ yields $(0.6, 1.0)$—indicating that the second concept attains its training-set maximum while the first is at $60\%$. The companion rescaling $W^* = S^{-1}\hat{W}$ keeps $UW$ invariant. In practice one may factor with $\hat{W}$ and normalize $U$ with $S$; both are equivalent. If a new sample exhibits an activation exceeding the training maximum (rare under i.i.d. settings), we clip it to $1$.

**Proposition 1** (Scale Equivalence). *For any diagonal matrix $S \succ 0$, the transform $(U, W) \mapsto (US, S^{-1}W)$ preserves reconstruction.*

*Proof.* We have $US \cdot S^{-1}W = U(SS^{-1})W = UW$, hence the reconstruction is invariant under the transform. $\square$

### 3.4 CLASS-SHARED CONCEPTS WITH CLASS-CONDITIONAL SPARSE GATING

To avoid redundant *per-class* concept learning and to improve cross-class reusability, we gate activations over a shared $W^*$ and introduce class-conditional gating parameters $\lambda \in \mathbb{R}^{c \times k}$. For $(a_i, y_i)$, let $g_{y_i} = \sigma(\lambda_{y_i}) \in (0,1)^k$, $\quad \hat{u}_i = u_i \odot g_{y_i}$, where $\lambda_{y_i}$ denotes the $y_i$-th row of $\lambda$; we then reconstruct $a_i \approx \hat{u}_i W^*$. A joint objective (alternating with Eq. (1)) is

$$U, \hat{W} = \min_{U \geq 0,\, W,\, \lambda} \frac{1}{n} \sum_{i=1}^n \left\| a_i - \left(u_i \cdot \mathrm{diag}(g_{y_i})\right) W \right\|_2^2. \tag{6}$$

**Optimization.** Given $(W, \lambda)$, we update $U$ with a few steps of *projected AdamW* (Loshchilov & Hutter, 2019) (AdamW step followed by projection onto the nonnegative orthant). We then update $W$ given $(U, \lambda)$, and update $\lambda$ given $(U, W)$ (backpropagating through $\sigma$).

**Proposition 2** (Two Equivalent Views of Gating). *For any $g \in [0,1]^k$, $(u \odot g)W = u \cdot (\mathrm{diag}(g)W)$.*

*Proof.* Gating on activations ($U$-side) is representationally equivalent to absorbing it into a class-specialized dictionary $W_y = \mathrm{diag}(g_y)W$. Training on the $U$-side stabilizes and preserves transferability of the shared $W$, while test-time interpretation may equivalently use $W_y$. $\square$

After training, we apply the normalization in §3.3 to obtain $(U, W^*)$, and fix $W^*$ for inference and downstream use.

**Inference-time Interaction.** Given a new feature $a \in \mathbb{R}^d$ (or batch $A_{\text{new}}$) and the fixed dictionary $W^*$, we solve the *nonnegative least squares* (NNLS) problem

$$u^* = \min_{u \geq 0} \|a - uW^*\|_2^2. \tag{7}$$

We approximate Eq. (7) with a few steps of *projected AdamW* (AdamW update followed by projection). NNLS may be used as an optional warm-start to initialize $u^{(0)}$; it is not on the critical optimization path.

**Proposition 3** (Properties of the NNLS Subproblem for Initialization; optional). *Although we do not rely on NNLS as the training/inference solver, when used for initialization with fixed $W^*$, the subproblem Eq. (7) is a convex quadratic program with an optimal solution. If the positive support $\mathcal{A}$ of an optimum satisfies that $W^*_{\mathcal{A}}$ has full column rank and strict complementarity holds, then the coefficient solution is unique.*

*Proof.* Define the linear image $\mathcal{C} := \{W^*u : u \geq 0\} \subset \mathbb{R}^d$. Since the nonnegative orthant $\mathbb{R}^k_{\geq 0}$ is a polyhedral cone, its image $\mathcal{C}$ under $u \mapsto W^*u$ is also a polyhedral cone and hence *closed and convex*. The problem

$$\min_{u \geq 0} \tfrac{1}{2}\|a - W^*u\|_2^2 \quad \Longleftrightarrow \quad \min_{y \in \mathcal{C}} \tfrac{1}{2}\|a - y\|_2^2 \tag{8}$$

is the Euclidean projection of $a$ onto a closed convex set, which admits a (unique) projected point $y^* = \mathrm{proj}_{\mathcal{C}}(a)$. Thus the NNLS optimum is attained by some $u^* \geq 0$ with $W^*u^* = y^*$.

For uniqueness of $u^*$, write the Lagrangian

$$L(u,\mu) = \tfrac{1}{2}\|a - W^*u\|_2^2 - \mu^\top u, \qquad \mu \geq 0, \tag{9}$$

with KKT conditions

$$(W^*)^\top(W^*u^* - a) - \mu^* = 0, \tag{10}$$

$$u^* \geq 0, \qquad \mu^* \geq 0, \tag{11}$$

$$\mu_i^* u_i^* = 0 \quad (\forall i). \tag{12}$$

Let the active (positive) support be $\mathcal{A} := \{i : u_i^* > 0\}$, $\mathcal{I} := \{i : u_i^* = 0\}$. Complementary slackness implies $\mu_{\mathcal{A}}^* = 0$ and $\mu_{\mathcal{I}}^* = (W_{\mathcal{I}}^*)^\top(W_{\mathcal{A}}^* u_{\mathcal{A}}^* - a) \geq 0$. Eq. (10) on the active block reduces to

$$(W_{\mathcal{A}}^*)^\top(W_{\mathcal{A}}^* u_{\mathcal{A}}^* - a) = 0. \tag{$\star$}$$

A sufficient nondegeneracy condition guaranteeing uniqueness is:

**(ND)** *Strict complementarity:* $u_{\mathcal{A}}^* > 0$, $u_{\mathcal{I}}^* = 0$, $\mu_{\mathcal{A}}^* = 0$, $\mu_{\mathcal{I}}^* > 0$; *Full column rank:* $W_{\mathcal{A}}^*$ has full column rank (equivalently, $(W_{\mathcal{A}}^*)^\top W_{\mathcal{A}}^* \succ 0$).

Under (ND), any feasible solution must share the same active set $\mathcal{A}$; otherwise complementary slackness is violated. With $\mathcal{A}$ fixed, the problem on the active face becomes the unconstrained quadratic

$$\min_{v \in \mathbb{R}^{|\mathcal{A}|}} \tfrac{1}{2}\|a - W_{\mathcal{A}}^* v\|_2^2, \tag{13}$$

whose first-order condition is ($\star$). Since $(W_{\mathcal{A}}^*)^\top W_{\mathcal{A}}^* \succ 0$, the unique solution is

$$u_{\mathcal{A}}^* = \left((W_{\mathcal{A}}^*)^\top W_{\mathcal{A}}^*\right)^{-1}(W_{\mathcal{A}}^*)^\top a, \tag{14}$$

together with $u_{\mathcal{I}}^* = 0$, yielding uniqueness of $u^*$. $\qquad\square$

## 3.5 Two Usage Modes for Concepts

After learning $(U, W^*)$ (and optional $g_y$), **SAND-CBM** provides two complementary, lightweight classification branches, each with a *single linear head* trained by cross-entropy.

**Branch-A: CBM style (text alignment + concept activations).** Construct the concatenated features

$$F_A = \begin{bmatrix} S \mid U \end{bmatrix} \in \mathbb{R}^{n \times (m+k)}, \qquad S = A_I \, E_T (P_T)^\top, \tag{15}$$

and predict with a linear classifier

$$\hat{y} = \mathrm{softmax}\big(\mathrm{Linear}(F_A)\big). \tag{16}$$

**Branch-B: CEM style (concept back-mapping + activations).** First map concepts back to CLIP space

$$Z = U \, W^* \in \mathbb{R}^{n \times d}, \tag{17}$$

and (in an "absorbed" implementation) fold the gating into the class-specialized linear head (or equivalently $W_y = \mathrm{diag}(g_y)W^*$), leaving the form of Eq. (18) unchanged; the explicit variant $[(U \odot g_y)W^* \mid U \odot g_y]$ is equivalent (Proposition 3). Concatenate with activations:

$$F_B = \begin{bmatrix} S \mid Z \end{bmatrix} \in \mathbb{R}^{n \times (m+d)}, \qquad \hat{y} = \mathrm{softmax}\big(\mathrm{Linear}(F_B)\big). \tag{18}$$

Given labels $\{y_i\}_{i=1}^n$, we train by cross-entropy loss ( (Shang et al., 2024; Oikarinen et al., 2023)).

## 4 Experiments

### 4.1 Experimental Setup

**Models and Datasets.** We evaluate on three benchmarks: **CIFAR–100** (Krizhevsky, 2009), **CUB** (Wah et al., 2011), and **SUN** (Xiao et al., 2010). We use 440 concepts on CIFAR–100, 312 concepts on CUB, and 102 concepts on SUN. Unless otherwise stated, all experiments adopt **ViT–B/16** (Dosovitskiy et al., 2021) as the backbone to ensure comparability.

**Hyperparameters.** Unless specified, we default to **Branch–B**. The number of concepts is set to $k = 200$. We use AdamW with learning rate $5 \times 10^{-3}$, weight decay $1 \times 10^{-4}$, batch size 128, and train for 1000 epochs, selecting the best checkpoint by validation **Top–1** accuracy. Experiments run on Red Hat Enterprise Linux 8.8 (kernel 4.18.0) with an Intel Xeon Gold 6442Y (24 cores, 2.60 GHz), 500 GB RAM, and two NVIDIA L40S GPUs (46 GB each), CUDA 12.6, driver 550.100. Software: Python 3.11, PyTorch 2.6.0, TorchVision 0.21.0.

**Baselines.** Beyond our **SAND–CBM**, we compare to four representative methods: **CF–CBM** (Dominici et al., 2024) (primary competitor), **LF–CBM** (Oikarinen et al., 2023) (no explicit concept labels), **PH–CBM** (Yuksekgonul et al., 2023) (post-hoc concept heads), and **Res–CBM** (Shang et al., 2024) (incremental residual completion). CF–CBM reports both high- and low-level concept results; we use their stronger variant.

**Metrics.** Top–$K$ accuracy is defined as $\text{Acc@}K = \frac{1}{N}\sum_{i=1}^{N} \mathbf{1}\big\{y_i \in \text{TopK}(p_i, K)\big\}$, where $\text{TopK}(p_i, K)$ returns the set of $K$ classes with the largest probabilities under $p_i$. In the main text we report **Acc@1**.

## 4.2 RESULTS

Table 1: Acc@1 on CIFAR–100, CUB, and SUN.

| Dataset | Res-CBM | LF-CBM | PH-CBM | CF-CBM | SAND-CBM |
|---|---|---|---|---|---|
| CIFAR-100 | **80.75** $(-0.23)$ | 76.58 $(+3.94)$ | 77.80 $(+2.72)$ | 77.13 $(+3.39)$ | 80.52 |
| CUB | 73.59 $(+7.17)$ | 68.76 $(+12.00)$ | 42.73 $(+38.03)$ | 77.20 $(+3.56)$ | **80.76** |
| SUN | 61.19 $(+6.45)$ | 48.95 $(+18.69)$ | 47.77 $(+19.87)$ | 61.51 $(+6.13)$ | **67.64** |

**Note.** Baseline columns show *"accuracy ($\Delta$)"*, where $\Delta = \text{Acc(SAND-CBM)} - \text{Acc(baseline)}$; the SAND-CBM column lists raw accuracies. Row-wise best is **bolded**. All values are percentages.

As shown in Tab. 1, across CIFAR–100, CUB, and SUN, **SAND–CBM** achieves consistently strong performance, improving on average over all baselines by $\approx 8.93$ and over the main competitor **CF–CBM** by $\approx 4.36$. On **CIFAR–100**, it reaches $80.52$, essentially on par with the strongest baseline Res–CBM $(80.75; -0.23)$ while improving over CF–CBM $(77.13; +3.39)$, LF–CBM $(76.58; +3.94)$, and PH–CBM $(77.80; +2.72)$; on **CUB**, it attains $80.76$, improving over Res–CBM by $+7.16$, over LF–CBM by $+12.00$, over PH–CBM by $+38.03$ (an $\approx 88.97\%$ relative gain), and over CF–CBM by $+3.56$; on **SUN**, it achieves $67.64$, surpassing Res–CBM by $+6.45$, LF–CBM by $+18.69$, PH–CBM by $+19.87$, and CF–CBM by $+6.14$, with gains over PH–CBM especially pronounced in scene recognition. Overall, **SAND–CBM** is state-of-the-art on fine-grained (CUB) and scene (SUN) classification while remaining competitive on generic object classification (CIFAR–100); the *macro-average* (equal-weight average across datasets) advantage over CF–CBM is $4.36$. Beyond accuracy, we further examine the structural properties of the learned dictionary: Appendix C reports a correlation analysis showing that most off-diagonal entries remain near zero, indicating low redundancy and well-separated concept directions, and we further visualize the learned class-conditional gating patterns in Appendix D; we complement the quantitative evaluation with a qualitative inspection of image concept heatmaps (Appendix E), which demonstrate consistent spatial footprints and clear between-concept separation, further validating the coherence and reusability of the learned concepts.

## 4.3 EFFECT OF THE DUAL-BRANCH MECHANISM

As shown in Fig. 2, three accuracy–$k$ curves reveal a clear division of labor. For *small concept budgets* ($k \lesssim 40$), **Branch-A** tends to win—its direct use of activations plus text alignment remains effective even with few concepts—whereas **Branch-B** has not yet formed strong high-dimensional compositions. As $k$ enters the *moderate regime* ($k \in [50, 120]$), Branch-B's advantage emerges: on CIFAR–100 and SUN it generally dominates for the same $k$ and exhibits smoother trends. For *large*

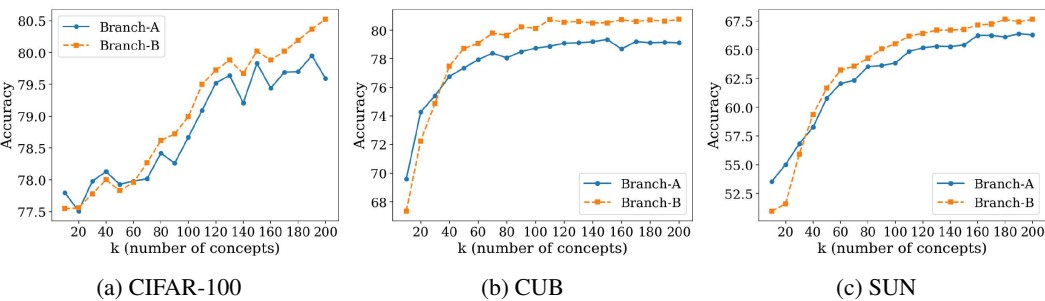

Figure 2: Accuracy vs. number of image concepts ($k$) across datasets, comparing Branch-A and Branch-B.

$k$ ($k \geq 100$), Branch-B reaches a higher ceiling: the best points on CIFAR–100 and SUN are from Branch-B; on CUB, Branch-B overtakes after $k \approx 100$. Intuitively, "back-mapping" ($Z = UW^*$) plus activations composes discriminative semantics more effectively when the concept library is rich, while Branch-A is more sensitive to noise amplification from text alignment and fluctuates at high $k$ on CIFAR–100. In practice, the branches are complementary: choose **A** for small $k$; choose **B** for higher ceilings once $k$ is sufficiently large. Our main results therefore report $k=200$.

### 4.4 ABLATION ON THE NUMBER OF IMAGE CONCEPTS $k$

As shown in Fig. 2, we vary $k \in \{10, 20, \ldots, 200\}$ on all three datasets while keeping other settings identical to the main protocol. *Overall trend.* Accuracy increases with $k$ for both branches with diminishing marginal returns; Branch-A is typically more robust at very small $k$ ($\leq 40$), while Branch-B tends to achieve the best final accuracy at larger $k$. *CIFAR–100.* Both curves rise with $k$. Branch-A shows visible fluctuations beyond $k \approx 130$; Branch-B is more monotonic and continues improving up to $k=200$, attaining the highest value. **Recommendation:** prefer Branch-B for mid-to-large $k$ (higher and smoother at equal $k$); Branch-A's instability past $k \gtrsim 125$ makes it a secondary choice unless text alignment is required. *CUB.* Both curves rise quickly and plateau. Branch-A saturates around $k \approx 110$ (near 79%), whereas Branch-B leads after $k > 40$ and still edges upward at $k=200$ (near 80.5%). For fine-grained data, enlarging the concept set yields early, substantial gains, followed by slower but steady improvements. *SUN.* Both curves increase steadily and flatten at high $k$. Branch-B starts lower at small $k$, surpasses Branch-A from mid $k$ onward, and plateaus around $k \approx 120$, finishing slightly higher (about 67.5% vs. 66%). This aligns with the higher semantic diversity of scenes. *Takeaways.* Larger $k$ raises the ceiling but with diminishing returns; Branch-B is generally superior and smoother in the mid-to-large $k$ regime, while Branch-A is competitive at small $k$. Visual "elbow points" are roughly: $\sim 130$ on CIFAR–100, $\sim 110$ on CUB, and $\sim 120$ on SUN. Practitioners may pick $k$ near these elbows for efficiency or push larger $k$ for marginal gains. We report $k=200$ for cross-dataset consistency and peak accuracy.

## 5 CONCLUSION

We presented **SAND-CBM**, a label-free concept bottleneck framework that discovers concepts from the image side by factorizing CLIP features into a *signed* dictionary $W$ and *nonnegative* activations $U$, followed by a scale-equivalent normalization that places each activation column in $[0, 1]$ for cross-concept comparability; a class-conditional sparse gate selects per-class subsets from a shared dictionary, enabling reuse without any per-class duplication. Built on the same $(U, W)$, two lightweight branches provide complementary usage modes: a CBM-style path that preserves compatibility with text-aligned interfaces, and a CEM-style path that back-maps concepts to the representation space to strengthen discriminative power. Experiments on CIFAR–100, CUB, and SUN show consistent gains over representative baselines, and diagnostics on dictionary correlation and gate patterns indicate that the model learns atomic, low-redundancy, and reusable directions that remain selective within each class.

ETHICS STATEMENT

We have read and will adhere to the ICLR Code of Ethics. This work uses only public data, involves no human subjects or personally identifiable information, and therefore does not require IRB review. Results are reported for research purposes only; we release anonymized code/configurations to support verification, and will disclose any funding sources and potential conflicts of interest upon acceptance.

REPRODUCIBILITY STATEMENT

To support reproducibility, we release an anonymized repository with all experiment details including training/evaluation scripts, default hyperparameters, configuration files, and software/hardware environment.

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

## LLM Usage Disclosure

We used large language models (OpenAI GPT-4o and GTP-5) as auxiliary tools for grammar checking and language polishing of the manuscript. These models were not involved in research ideation, experimental design, implementation, or analysis. The authors take full responsibility for all content.

## A  Design and Implementation Notes and Theoretical Summary

**SAND-CBM** applies a linear, interpretable factorization to CLIP features: we constrain only the activations ($U \geq 0$) while allowing a *signed* dictionary $W$ in CLIP's semantic space. This preserves the interpretability of *additive strengths* (via $U$) without violating CLIP's intrinsic sign structure (unlike NMF's $W \geq 0$). Thus "concept strength" is read directly from $U$, while "concept direction" is encoded by $W$.

For cross-concept comparability, we enforce the *scale-equivalent* transform $(U, \hat{W}) \mapsto (US, S^{-1}\hat{W})$ (Eq. (4)) to map each activation column to $[0, 1]$. Proposition 1 guarantees that normalization preserves the optimal reconstruction; test-time $u^*$ can therefore be interpreted on a *normalized* strength scale established during training.

To balance sharing and specificity, we place class selectivity on the activation side via $g_y = \sigma(\lambda_y)$ (§3.4), stabilizing and reusing a shared $W$. Proposition 3 shows that gating can be equivalently absorbed into a class-specialized dictionary $W_y = \mathrm{diag}(g_y)W$, allowing flexibility in training and visualization.

Implementation-wise, we use a small number of projected-AdamW steps as the main workhorse: iteratively update $U$ in Eq. (1) with AdamW followed by projection to the nonnegative orthant; at inference, solve the fixed-$W^*$ NNLS in Eq. (7) with the same scheme. *NNLS is optional and used only as a warm start.*

Theoretically, Proposition 1 establishes scale equivalence; Proposition 2 formalizes the equivalence between activation-side gating and dictionary-side absorption; Proposition 3 shows that the fixed-$W^*$ subproblem is a convex QP with an attained optimum and, under standard nondegeneracy (strict complementarity + full column rank on the active subdictionary), a unique coefficient solution. These properties justify our choices in Eq. (1), Eq.(4), and Eq.(7) and support an efficient, stable implementation.

## B  Computational Complexity.

For the base factorization in Eq. (1), a single optimization step that updates $U$ and $W$ (i.e., one forward reconstruction $UW$ together with the associated gradient computations) costs $\mathcal{O}(nkd)$. Over $T$ NMF-style (projected) optimization steps, the total training complexity is $\mathcal{O}(Tnkd)$. Here, $n$ denotes the number of training samples, $k$ the number of learned image concepts, $d$ the CLIP feature dimension, and $T$ the number of optimization steps. The projection onto the nonnegative orthant for $U$ adds negligible overhead compared with the matrix multiplications.

## C  Diagnosing De-correlation in the Concept Dictionary

To evaluate the separation and complementarity of learned concept directions, we compute the Pearson correlation matrix of the dictionary rows $R = \mathrm{corrcoef}(W) \in \mathbb{R}^{k \times k}$ and use

$$\mathrm{OffDiagCorr} \;=\; \frac{\|R - \mathrm{diag}(R)\|_F^2}{k(k-1)}$$

as a de-correlation metric (mean squared off-diagonal correlation; lower is better). With $k{=}10$, we obtain: SUN 0.0136, CIFAR–100 0.0334, CUB 0.0377 (Fig. 3c–3b). The majority of off-diagonal entries remain close to zero, suggesting well-separated concept directions and reduced redundancy—consistent with our *nonnegative activations $U$* (strength) plus *signed dictionary $W$* (direction) design in §3.3, which promotes additive, compositional representations. CUB exhibits moderately higher correlations (e.g., color–part co-occurrences), which may reflect meaningful

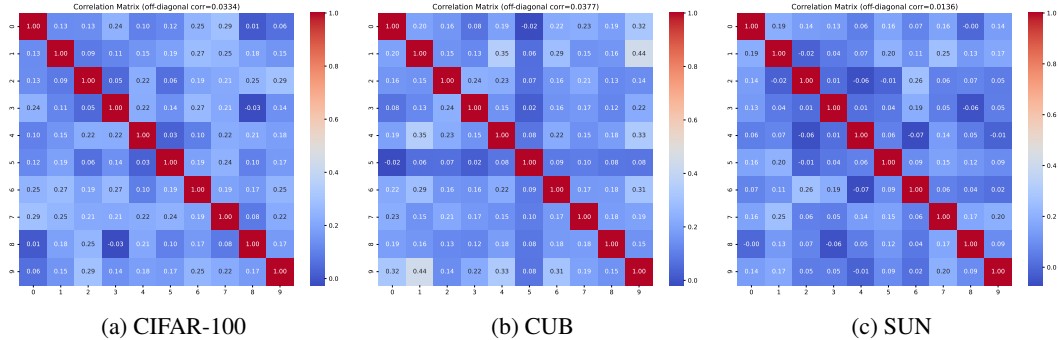

(a) CIFAR-100        (b) CUB        (c) SUN

Figure 3: Row-wise correlation matrices of the learned dictionary $W$ at $k=10$. OffDiagCorr is the mean squared off-diagonal correlation (lower is better). Most off-diagonal entries are near zero, indicating separated directions and low redundancy.



(a) CUB        (b) CIFAR-100        (c) SUN

Figure 4: Class-conditional gate heatmaps ($k=10$). Columns: classes; rows: concepts. Values are post-softmax weights over concepts (brighter is larger). x–axis: class id; y–axis: concept id.

semantic relationships rather than dictionary collapse; the class-conditional gating in §3.4 can leverage such co-occurrences as effective per-class subsets at inference. Notably, these low correlations emerge *without* explicit orthogonality constraints, suggesting that SAND–CBM tends to discover complementary directions—supporting the empirical observations in §3.5 and §3.3 that increasing $k$ expands representational capacity rather than duplicating concepts.

## D  VISUALIZING CLASS-CONDITIONAL GATING ($k=10$)

We apply a softmax over the concept dimension of the gating matrix and visualize the transpose as a $k \times c$ heatmap. As shown in Fig. 4 (CUB, CIFAR–100, SUN in Subfigs. a, b, c), all three datasets demonstrate *sparse-within-class, shared-across-classes* activation patterns: individual classes activate concentrated subsets of concepts while the same concepts participate across multiple classes. The selectivity patterns vary across datasets—with sharper peaks on CUB, moderate selectivity on CIFAR–100, and smoother distributions on SUN—potentially reflecting different levels of inter-class semantic overlap. These patterns align with our "shared $W$ + class-conditional $\sigma(\lambda_y)$" design and, combined with the directional separation observed in §C, provide evidence that the learned representations exhibit both specialization and reusability properties.

## E  QUALITATIVE ANALYSIS OF IMAGE CONCEPTS AND HEATMAPS

**What the heatmaps show.** Each heatmap visualizes the *spatial support* of a learned image concept: brighter regions indicate where the concept's activation contributes most to the reconstruction or to the downstream decision. This makes the heatmap a localization cue—useful for seeing *which* parts of the image the model attends to for a given concept.

**What the heatmaps do *not* show.** A heatmap does *not* reveal the concept's semantic content by itself. It cannot disambiguate whether a highlighted region corresponds to color, texture, shape, or a conjunction of attributes. Thus, heatmaps indicate *where* but not *what*; they should be read as spatial footprints rather than direct concept definitions.

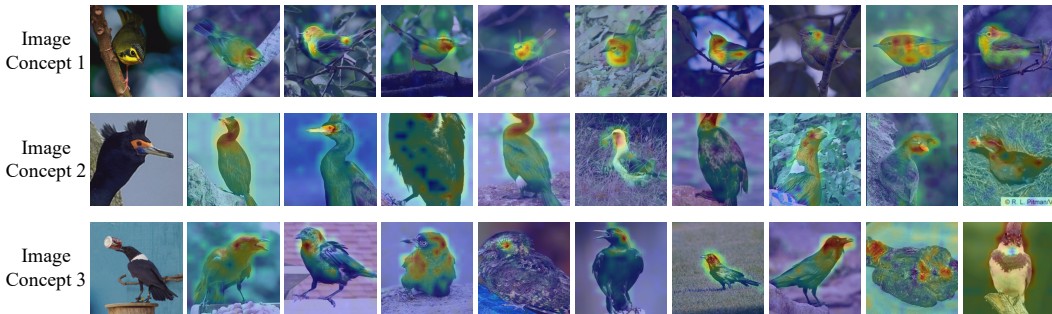

Figure 5: Examples of image concepts with corresponding heatmaps. Brighter regions indicate higher concept activation. Heatmaps localize *where* the model attends but do not specify the concept's semantic content, so they are best used to assess cross-image consistency of each learned concept.

**Consistency check for learned concepts.** Despite this limitation, heatmaps are effective for diagnosing *concept consistency*. For a fixed concept, we expect similar spatial patterns across images that truly instantiate that concept (e.g., consistent focus on beaks for bird-related concepts or on horizon bands for scene-related concepts). Visual inspection across multiple images helps identify whether a concept has stable localization or is diffuse and class-agnostic. In practice, we (i) inspect per-concept grids of images with their heatmaps, (ii) look for tight, repeatable regions, and (iii) flag concepts whose activations migrate unpredictably. Consistent spatial footprints suggest that the learned image concept is coherent; inconsistent or scattered footprints suggest entanglement or spurious correlations and warrant re-training or adjustment of the concept budget and gating strength.

**Detailed analysis of the heatmaps in Fig. 5.** **Image Concept 1** shows a compact and repeatable hotspot on the anterior head region across almost all images. The peak typically sits around the eye stripe and crown, remains stable under pose changes, and only slightly spills to the shoulder when the head is partially occluded. Background foliage and branches are largely suppressed. **Image Concept 2** concentrates on an elongated band that tracks the neck and the base of the bill. The activation extends vertically along the throat and upper breast for upright perching poses, while torso and background remain cool. This pattern holds across side views and three-quarter views, indicating a consistent spatial footprint tied to the neck column rather than the whole body. **Image Concept 3** focuses on a small region near the bill tip and gape, often when the bird is vocalizing or the mouth is open. The hotspot stays anchored at the head centroid even under clutter and nonstandard crops, with minimal leakage to wings or background.

Taken together, the three rows exhibit *high within-concept consistency*—each concept returns to a similar, compact locus across many instances—and *clear between-concept separation*—head patch versus neck column versus bill tip—resulting in limited overlap among their spatial supports. The heatmaps indicate *where* each concept contributes rather than *what* the concept means; nonetheless, the tight and repeatable footprints provide qualitative evidence that the learned image concepts are coherent and reusable. This observation aligns with the low inter-row correlation of dictionary directions reported in § C.

