# OpenReview forum: "Signed-dictionary and Nonnegative-activation Decomposition for Concept Bottleneck Models"
_ICLR.cc/2026/Conference — ICLR 2026 Conference Withdrawn Submission_

### Official Review · Reviewer_6eeN · 2025-10-29

**Soundness:** 2
**Presentation:** 2
**Contribution:** 3
**Rating:** 2
**Confidence:** 4

**Summary:**

This paper aims to overcome the limitations of label-free CBMs, that rely heavily on incomplete or biased text pools for concept discovery. To this end, the authors propose SAND-CBM, a novel framework that directly learns image-side concepts in a vision–language aligned space. SAND-CBM factorizes CLIP features into a signed concept dictionary and nonnegative activations, followed by scale-equivalent normalization and class-conditional sparse gating to enable shared yet selective concept usage across classes. Through extensive experiments the authors demonstrate that the classification accuracy of SAND-CBM consistently surpasses SOTA baselines.

**Strengths:**

* This paper makes the first attempt to construct concepts directly from image features for the label-free CBM problem, effectively avoiding issues such as text bias and incomplete semantic coverage.
* The authors discuss the related literature in considerable detail.
* The paper is well written and easy to follow.
* The authors provide code to facilitate reproducibility checks.

**Weaknesses:**

1. **Interpretability**: As an interpretable model architecture, I have concerns regarding the interpretability of the proposed method. Unlike traditional concepts with explicit semantic correspondence, the concepts are directly extracted from image features, leading to complex and coupled semantics. More importantly, they lack interpretability, making these "concepts" closer to uninterpretable deep features. However, in the experimental section, the paper only compares classification accuracy across methods. While the proposed approach achieves significant performance gains, it lacks sufficient qualitative and quantitative analyses of interpretability. The authors should further explain the extracted concepts, including semantic decomposition, evaluation of semantic consistency (whether the same concept corresponds to semantically consistent regions across images), and the extent of concept reuse.
2. **Method**: The paper employs the CLIP model to extract aligned visual and textual features. However, in pre-training stage, CLIP only aligns global image features with semantic representations, without explicit supervision on local semantics. Would this limitation affect the semantic interpretability of the concepts constructed by the proposed method?
3. The text in Figure 1 is too small, making it difficult to read, and the distinction between Branch A and Branch B is not visually clear.

**Questions:**

My questions are in Weaknesses Section.

---

### Official Review · Reviewer_FFV8 · 2025-10-30

**Soundness:** 2
**Presentation:** 3
**Contribution:** 2
**Rating:** 4
**Confidence:** 3

**Summary:**

The paper introduces \textbf{SAND-CBM}, a label-free framework for Concept Bottleneck Models that learns concepts directly from image representations in an aligned vision-language space, addressing the limitations of fixed, predefined concept pools. The core technique is feature factorization into a \textit{Signed Concept Dictionary} ($W$) and \textit{Nonnegative Activations} ($U$), coupled with a scale-equivalent normalization. A class-conditional sparse gate is used to select and reuse a shared dictionary across classes. SAND-CBM exposes two complementary usage modes: a CBM-style interface (Branch-A) and a CEM-style interface (Branch-B).

**Strengths:**

1. This work enables label-free concept discovery directly from data in the V-L space, significantly broadening the scope beyond fixed, text-based concept pools.

2. The dictionary decomposition ($W$ signed, $U$ nonnegative) promotes robust and reusable concept learning, as qualitatively evidenced by the consistent and spatially separated concept footprints.

3. The framework is flexible, offering interpretability interfaces in CBM-style (Branch-A) and CEM-style (Branch-B) using the same core components ($U, W$).

**Weaknesses:**

The label-free concept discovery relies on an *aligned vision-language space*, which introduces a strong dependency on the quality and biases of the underlying pre-trained V-L model. The necessity of two distinct usage modes (Branch-A and Branch-B) implies that neither is universally optimal, potentially complicating the choice of interface for downstream users.

**Questions:**

Since the paper aims for interpretable concepts, what measures or metrics (beyond spatial consistency) are used to quantitatively verify the \textit{human-understandability} or fidelity of the discovered concepts in $W$ and $U$, especially in comparison to concepts explicitly defined by human labels in traditional CBMs?

---

### Official Review · Reviewer_D2cd · 2025-10-30

**Soundness:** 3
**Presentation:** 3
**Contribution:** 2
**Rating:** 2
**Confidence:** 4

**Summary:**

This paper proposes a Concept Bottleneck Model (CBM) framework that learns fine-grained concepts directly from image representations and does not rely on text concepts which don’t exhaustively capture fine-grained and compositional concepts.

**Strengths:**

- The authors propose a dictionary learning objective for learning non-negative and normalized activation values for concepts directly from image representations

- Proposed pipeline and learned dictionary can be used in both CBM and CEM manner

**Weaknesses:**

- Previous work including[3] have explored similar dictionary learning strategies for Concept Bottleneck Models and this work is missing comparison with these baselines. Further, this work is also missing comparison with recent baselines including [1,4].
- This work is missing results on larger datasets, including Imagenet and Places365, following [2,4]. These results are important to understand the scalability of the proposed approach as dataset size increases.
- Recent studies [3, 4] have shown that CBM training can leak information through the bottleneck layer, allowing models to achieve high accuracy without actually learning meaningful concepts. To address this issue, [4] introduced the A-NEC metric to enable fair comparisons between CBMs based on their concept learning ability and SAND-CBM should be evaluated using this metric for a fair comparison. The manuscript also has very limited visualization of the concepts learned by method.
- One of the primary motivations for CBM is to use explanations for downstream model editing and intervention. However, the lack of grounding to text concepts makes these tasks very challenging. The authors should discuss application scenarios for the proposed approach.


[1] Yang, Yue, et al. "Language in a bottle: Language model guided concept bottlenecks for interpretable image classification." Proceedings of the IEEE/CVF conference on computer vision and pattern recognition. 2023.

[2] Oikarinen, Tuomas, et al. "Label-free concept bottleneck models." arXiv preprint arXiv:2304.06129 (2023).

[3] Yan, An, et al. "Learning concise and descriptive attributes for visual recognition." Proceedings of the IEEE/CVF International Conference on Computer Vision. 2023.

[4] Srivastava, Divyansh, Ge Yan, and Lily Weng. "Vlg-cbm: Training concept bottleneck models with vision-language guidance." Advances in Neural Information Processing Systems 37 (2024): 79057-79094.

**Questions:**

My primary concerns are limited baselines, lack of results on larger datasets including ImageNet and Places365, and missing metrics which compare CBMs fairly in terms of their concept learning capability. Further, the authors should provide additional qualitative results demonstrating concepts learned for different classes and datasets. Please see weaknesses for more details.

---

### Official Review · Reviewer_rSeX · 2025-10-31

**Soundness:** 3
**Presentation:** 2
**Contribution:** 2
**Rating:** 4
**Confidence:** 4

**Summary:**

The authors develop a two-branched system that allows for not relying solely on textual concepts but also producing them on the fly using a pretrained CLIP's image encoder and back-mapped concept reconstructions. Such an approach increases the expressivity of the model since it now can leverage textual concepts (signed dictionary of concepts) that are more tailored to images and their classes than the basic related words obtained by the common knowledge of human experts, from LLMs, etc.
The dual-branch mechanism, where one branch relies on predefined concepts and another that learns a new set of concepts and activations in an aligned semantic (image-language) space, allows for using the best of both worlds --- a strong explanatory ability and discriminative power. With this framework, the authors demonstrate highly competitive performance on popular community tasks across various concept budgets.

**Strengths:**

**Well-explained framework and theoretical statements.** Explanation and statements in Sec. 3.3 are sound and novel in the field. The provided propositions show the feasibility of the authors' ideas. However, the details of the proof of Proposition 3 can be moved to Appendix; instead, one can provide more experiments underscoring the intuition of the framework, showing the process of learning the concept dictionary from the image encoder.

**Clear codebase.** Code repository is clean and organized properly. Figure in the `README.md` summarizes author's framework very well.

**Dual-branched mechanism.** The idea of not relying too much on the predefined set of concepts --- that is typically hand-crafted or provided by LLM --- is relatively novel and worth potential investigation in the future research.

**Weaknesses:**

**Incompleteness and mismatch in Experiments.**
Despite being well-motivated and theoretically discussed with many details in Sec. 3.3-3.4, the experimental part is vague and lacks an important discussions.
1) Since the number of concepts (k) is a hyperparameter, it is important to show how the optimization problem (Eq. 1, Eq. 6) complicates with k or show that this is not significant to compute the solution. Also, the authors mention running AdamW for a *few iterations* (Line 293). For how many steps do you run it? Can you formalize the precise procedure in Appendix? I think this is curial because the same procedure is also used during the inference (Line 309).
2) Mismatch with the codebase. In Line 377, you write that using 312 concepts for CUB. However, in your repo, I see 370 concepts. Correct, please.
3) Why do you show present ablations in Fig. 2 with only up to 200 concepts per dataset? Ablations with k = 440 for CIFAR, or k = 370 for CUB are missing. Why not using them in Fig. 2 if you mention their quantity?
4) The analysis of plots in Fig. 2, provided in Sec. 4.4, is very detailed. But, in my opinion, the main aspect is missing --- different datasets require different number of concepts --- this is clearly evident from the huge discrepancy in the metrics in Table 1 --- consequently, it worth considering different number of concepts for different datasets. Moreover, this aligns with prior finding of your baselines, since LF-CBM, PH-CBM, Res-CBM, CF-CBM all use different number of concepts w.r.t. dataset. For example, in LF-CBM work, the authors use 211 concepts for CUB, and 824 for CIFAR-100 --- a way more than you do. For ImageNet, this value even exceeds 4k concepts. On the other hand, the Res-CBM work notices that the number of concepts should not be the larger the better, meaning there is some critical k, beyond which, the accuracy will drop. It would be interesting to see experiments showing this in your experiments as well.
5) Regarding hyperparameters. From the code, I see that you are using weight decay of 0.0001, I recommend using larger values, e.g., 0.1 because you use the decoupled weight decay strategy.
6) Weird number is Table 1. For baselines, you report values that dramatically differ from ones in the original works. E.g., for LF-CBM you report 76.58% on CIFAR-100 and 68.76% on CUB, while in the original work those are ~65% on CIFAR-100 and ~74% on CUB. The same for PH-CBM, yours (77.80%, 42.73%) vs. original (52%, 58.8%) on CIFAR-100 and CUB, resp. As an additional reference, the LF-CBM work reports the following accuracies for PH-CBM on CIFAR-100 and CUB (43.20%, 59.60%). Could you explain the difference, please? Because it seems that both baselines use the same ViT-B/16 backbone image encoder.

**Limited baseline and datasets.**
1) Assuming that previous issue is resolved, your framework does fairly well in Table 1. But there are still missing baselines to add, e.g., [1, 2, 3, 4]. All works provide community with new solutions and claim to outperform the long-standing LF- and PH- CBMs baselines with quite a margin. Including necessary comparisons or, at least discussing potential strong baselines, would make the work more comprehensive and appealing.
2) In your code, I see the option for Places365 and ImageNet dataset, however, those are not employed in the submission manuscript. I recommend authors to try to add them because those are important problems for comparison and are using by many competitive baselines you citing and missing. I also recommend authors to try a simple problem --- CIFAR10, to see wether your method, which solves Eq. 6 and alike, can do very well (91%+) on this data.

**No examples with concepts extraction.** I would like to see which textual concepts does your  model recover while doing classification. There are many examples of this kind in concurrent and prior works, so omitting this significantly undermines your contribution.

**Error in references.**  When referring to CF-CBM, you are citing the "Coarse-to-fine concept bottleneck models" 2024 paper. However, this paper is authored by Konstantinos P. Panousis and not by Elisa Dominici. Moreover, Elisa Dominici is not the author of any paper on CBMs! Meanwhile, another concurrent the "Counterfactual Concept Bottleneck Models" 2024 (accepted in 2025) paper --- which is also CF-CBM --- is authored by Gabriele Dominici. Please, correct your mistake in references.



[1]  "VLG-CBM: Training Concept Bottleneck Models with Vision-Language Guidance", 2024

[2] "The Decoupling Concept Bottleneck Model", 2024

[3] "Sparse Concept Bottleneck Models: Gumbel Tricks in Contrastive Learning", 2024

[4] "Improving Concept Alignment in Vision-Language Concept Bottleneck Models", 2024

**Questions:**

See the **Weaknesses** part

**Details Of Ethics Concerns:**

I have found no ethics concerns.
All the data is presented in the code base, and not a single person participated in the creation of a set of concepts.
These sets of concepts are also harmless.

---

### Note · Authors · 2025-11-14

I have read and agree with the venue's withdrawal policy on behalf of myself and my co-authors.